# iTRAQ-Based Proteomics Analysis of Autophagy-Mediated Responses against MeJA in Laticifers of *Euphorbia kansui* L.

**DOI:** 10.3390/ijms20153770

**Published:** 2019-08-01

**Authors:** Xiaoai Fang, Xiangyu Yao, Yue Zhang, Zheni Tian, Meng Wang, Peng Li, Xia Cai

**Affiliations:** Key Laboratory of Resource Biology and Biotechnology in Western China, Ministry of Education, Northwest University, Xi’an 710069, China

**Keywords:** methyl jasmonate, iTRAQ, ATG8, ATG18a, autophagy, ROS, *Euphorbia kansui*

## Abstract

Autophagy is a well-defined catabolic mechanism whereby cytoplasmic materials are engulfed into a structure termed the autophagosome. Methyl jasmonate (MeJA), a plant hormone, mediates diverse developmental process and defense responses which induce a variety of metabolites. In plants, little is known about autophagy-mediated responses against MeJA. In this study, we used high-throughput comparative proteomics to identify proteins of latex in the laticifers. The isobaric tags for relative and absolute quantification (iTRAQ) MS/MS proteomics were performed, and 298 proteins among MeJA treated groups and the control group of *Euphorbia kansui* were identified. It is interesting to note that 29 significant differentially expressed proteins were identified and their associations with autophagy and ROS pathway were verified for several selected proteins as follows: α-L-fucosidase, β-galactosidase, cysteine proteinase, and Cu/Zn superoxide dismutase. Quantitative real-time PCR analysis of the selected genes confirmed the fact that MeJA might enhance the expression of some genes related to autophagy. The western blotting and immunofluorescence results of ATG8 and ATG18a which are two important proteins for the formation of autophagosomes also demonstrated that MeJA could promote autophagy at the protein level. Using the electron microscope, we observed an increase in autophagosomes after MeJA treatment. These results indicated that MeJA might promote autophagy in *E. kansui* laticifers; and it was speculated that MeJA mediated autophagy through two possible ways: the increase of ROS induces ATG8 accumulation and then aotophagosome formation, and MeJA promotes ATG18 accumulation and then autophagosome formation. Taken together, our results provide several novel insights for understanding the mechanism between autophagy and MeJA treatment. However, the specific mechanism remains to be further studied in the future.

## 1. Introduction

Jasmonate (JA) and its volatile form, methyl jasmonate (MeJA) are collectively called jasmonates (JAs). As a type of plant hormones, JAs involved in regulating diverse development processes (fertility, seed germination, root growth, and fruit ripening) and defense responses to abiotic (drought, low temperature, and salinity) and biotic stresses (insect-driven wounding and various pathogens) [1,2,3]. Over the past decade, evidences that exogenous application of JAs on plants can affect various stages of plant physiological processes have been accumulating [1,2,3,4]. For example, MeJA aggravated growth inhibition and senescence in *Artemisia absinthium* L. [5] and *Arabidopsis* [6,7], its role in senescence is linked to the down-regulation of housekeeping proteins encoded by photosynthetic genes and the up-regulation of genes active in reactions against stress [8,9,10]. MeJA promotes the transcription and expression of the vegetative storage protein (VSP) in *Medicago sativa* [11], and also induces the biosynthesis of secondary plant metabolites. It has been reported that exogenous JAs can stimulate formation and differentiation of laticifers, a specific type of cell, in *Hevea brasiliensis*, and they can induce the production of numerous secondary metabolities, including latex [11,12].

Autophagy, a life-promoting lysosomal degradation pathway is conservative in eukaryotics [13,14]. Autophagy, which means “self-eating,” is a protein degradation process in which cells recycle cytoplasmic contents when subjected to environmental stress conditions or during certain stages of development [15]. This process mediates the degradation and recycling of cellular components through their segregation into double-membrane vesicles called autophagosomes, which will deliver the contents to lysosomes or vacuole for degradation by hydrolases. The functional relationship between apoptosis and autophagy is complex in the sense that, under certain circumstances, autophagy constitutes a stress adaptation that suppresses apoptosis, whereas in other cellular settings, it constitutes an alternative cell-death pathway [15]. In plant, the importance of autophagy has generated significant interest in various biological processes during normal growth and development, for example, protein aggregates are transported to vacuoles by a macroautophagic mechanism in nutrient-starved plant cells; autophagy maintains the metabolism and function of young and old stem cells [16,17]. Autophagy is essential for the degradation of oxidized proteins during oxidative stress in plants [18,19], for the degradation of aggregated proteins in nutrient-starved plant cells [15] and for limiting the cell death response during pathogen attack [20,21,22]. Numerous researches have been reported to elucidate that the MeJA can induce autophagy in cancer, for example, MeJA induces p53-dependent apoptotic and p53-independent non-apoptotic cell death in cancer [23,24,25], MeJA induces apoptosis via the ROS pathway, while simultaneously inducing pro-apoptotic autophagy in human non-small cell lung cancer [23]. However, few studies have been explored to elucidate the effect of MeJA on autophagy in plant, whether it stimulates autophagy via ROS pathway like mammalians, or promote autophagy directly? This problem needs to be solved.

*Euphorbia kansui* L. is a medicinal plant belonging to the Euphorbiaceae family, which is well known for its milky latex substance. The latex occurs within specialized secretory cells that are called as laticifers. The occurrence of non-articulated laticifers has been noted in the literatures for the Euphorbiaceae family. In various species, laticifers are mainly the synthesis and storage sites of secondary metabolites, which have important economic value. The research of laticifer development is of great significance to further study the regulation and synthesis mechanism of important secondary metabolites. The non-articulated laticifers develop from a single primary cell; as a result, the latex is its protoplasm apart from the cell wall and plasma membrane. Therefore, latex from non-articulated laticifers in *E. kansui* is the best material for proteomic research to analyses the laticifer development and secondary metabolite synthesis. In our previous reports, we found that the lysosomal pathway existed in the developmental laticifers of *E. helioscopia* L., in addition, autophagy was involved in the development process and the maintenance of intracellular homeostasis of the laticifers in *E. kansui* [26,27].

In this study, iTRAQ coupled with MS/MS technique was applied for the proteomic analysis of *Euphorbia kansui* to identify changes of laticifer proteins to MeJA treatment. Combining the de novo annotation of *E. kansui* transcriptomic data with iTRAQ analysis; and on the basis of gene ontology (GO) and Kyoto Encyclopedia of Genes and Genomes (KEGG) analysis, we identified some proteins showing significant differences were related to ROS pathway and autophagy. The transcript levels of selected proteins were investigated with real-time PCR. We also verified the effect of MeJA on autophagy related proteins with the methods of western blotting and immunofluorescence. Finally, we observed the ultrastructural changes of *E. kansui* laticifers treated with MeJA. These results reveal that MeJA promote the autophagy in *E. kansui* laticifers. According our results and references, a hypothetical model was proposed: MeJA treatment increases ROS production, which induces ATG8 accumulation and then aotophagosome formation, and MeJA promotes ATG18 accumulation and then autophagosome formation.

## 2. Results

### 2.1. Primary Data Analysis and Protein Identification

In this study, iTRAQ (isobaric tags for relative and absolute quantification) was used to assess proteome changes in *E. kansui* that treated with MeJA and the control group. In total, 509 proteins were acquired from the NCBI Sequence Read Archive database on the basis of mRNA transcriptome of *Euphorbia kansui* in response to MeJA (Accession number: SRP126436) with ProteinPilot 5.0 software, including 305, 172, and 32 proteins with a mass of 10–50 kDa, 50–100 kDa, and more than 100 kDa, respectively (Figure 1A), of which, 297 proteins were annotated (Appendix A), and 29 significantly differentially expressed proteins were identified (Appendix A). The distribution of peptide numbers is shown in Figure 1B, the proteins with a single peptide, 2–5 peptides, 6–10 peptides, and ≥11 peptides consisted of 257, 188, 39, and 25 respectively. The number of proteins with sequence coverage of ≤ 5%, 5%–15%, 15%–30%, 30%–100% were respectively 197, 207, 77, and 28 (Figure 1C).

### 2.2. Gene Ontology Analysis

Predicted molecular functions were assigned to proteins identified using the functional annotation tool for gene ontology (GO) term enrichment analysis. In total, 29 differentially expressed proteins were identified in MeJA treated groups VS control groups. The GO analysis for all differentially expressed proteins is shown in Figure 2. A total of 16 biological processes, 9 cellular components, and 8 molecular functions were affected among all samples by MeJA (Figure 2).

### 2.3. KEGG Enriched Pathways

To further classify functional annotations of these proteins, the Kyoto Encyclopedia of Genes and Genomes (KEGG) ontology assignments were carried out. According to our work, top 20 KEGG pathways were enriched (Figure 3, Appendix A). Among these pathways, 3 significantly enriched pathways were identified when *p*-value was less than 0.05, including Nicotinate and nicotinamide metabolism (3 proteins, 10.34%), Folate biosynthesis (2 proteins, 6.9%), and Terpenoid backbone biosynthesis (2 proteins, 6.9%). Among the differentially expressed proteins, metabolic pathways were the most enriched, including Thiamine metabolism, Spliceosomem One carbon pool by folate, Glutathione metabolism, Glyoxylate and dicarboxylate metabolism, mRNA surveilance pathway, SNARE interactions in vesicular transport, Cyanoamino acid metabolism, Plant-pathogen interaction, Ubiquinone and the other terpeniod-quinone biosynthesis, Porphyrin and chlorophyll metabolism, Purine metabolism, Glycine, serine and threonine metabolism, Alanine, aspartate and glutamate metabolism, RNA transport, RNA transport, Aminoacyl-tRNA biosynthesis, and Plant hormone signal transduction.

### 2.4. Quantitative RT-PCR (qRT-PCR) Analysis on Expression of Genes of Significantly Differentially Expressed Proteins under MeJA Stress

We validated several genes closely associated with MeJA stress on autophagy for qRT-PCR analysis: genes of three important components of lysosome including α-L-fucosidase, β-galactosidase, cysteine proteinase, a gene of Cu/Zn superoxide dismutase which is a ROS pathway protein, 6 genes of autophagy related proteins (ATG protein). The expression of these genes was analyzed by RT-qPCR using mRNA of latex of 3-month-old *E. kansui* wild-type plants. The primers of these genes and reference gene (actin) are listed in Appendix A. The qRT-PCR results were shown in Figure 4A,B. Following MeJA treatment 36 h (MeJA36), there was more than 2.0-fold increase in the expression levels of these genes including β-galactosidase, cysteine proteinase, Cu/Zn superoxide dismutase. However, α-L-fucosidase was continuously up-regulated under MeJA treatment 48 h (MeJA48). These results were consistent with the proteome results. Genes of ATG1c, ATG3, ATG7, ATG8s, and ATG18s were almost up-regulated significantly at MeJA treatment 36 h (MeJA36) (*p*-value ≤ 0.01) and then down-regulated at MeJA treatment 48 h (MeJA48). Therefore, MeJA could promote the expression of genes associated with autophagy in *Euphorbia kansui* laticifers.

### 2.5. Immunoblotting Analysis of ATG8 and ATG18a in Laticifers under MeJA Treatment

The autophagy related protein 8 (ATG8) is located on the autophagosome membrane and is often used as autophagosome marker in eukaryotic cells [28]. ATG8, whose activity was found to be essential for autophagosome formation, regulates membrane elongation during autophagosome biogenesis [29]. The autophagy related protein 18 (ATG18) plays a role in the initial stage of autophagosome formation. After the cells receive the signal to initiate the autophagy process, phosphatidylinositol 3-kinase (PI3K) is responsible for the synthesis of phosphatidylinositol 3-phosphate, 1, 2-dipalmitoyl (PI3P) at the initial site of pre-autophagosome (PAS) membrane. ATG18a is located on the surface of autophagosome membrane by binding to PI3P and interacts with autophagy related protein 2 (ATG2) to form atg2-atg18a complex, whose role is circulating autophagy related protein 9 (ATG9) to transport lipids through PAS [29,30]. Therefore, both ATG8 and ATG18a could be used to label the autophagosome structure.

To further verify that MeJA might promote autophagy in laticifers, we performed western blot analysis with ATG8 and ATG18a. Antibodies against ATG18a were generated with 6×His recombinant protein as antigen. Recombinant 6 × His-ATG18a proteins were purified from insoluble fractions as His fusion proteins using a Protein A column according to standard methods (GE) The sequence information of genes was in Appendix A.

We extracted total proteins of latex from the control group, MeJA treatment 36 h (MeJA36) group and MeJA treatment 48 h (MeJA48) group and detected with antibodies against ATG8, ATG18a respectively. Coomassie stained gel image showed equal loading of protein samples (20 μg each lane) (Figure 5A). Western-blotting grayscale value analysis revealed that there showed 6.30-fold and 1.44-fold increase of ATG8a abundance (Figure 5B,D), 2.61-fold and 2.06-fold increase of ATG18a abundance in MeJA treatment 36 h (MeJA36) group (*p*-value ≤ 0.01) and MeJA treatment 48 h (MeJA48) group compared to the control group (Figure 5C,E), respectively.

### 2.6. Confocal Immunofluorescence Analysis and TEM Analysis of Autophagy in Laticifers with MeJA Treatment

Monodansylcadaverine (MDC), a fluorescent drug, was used as a probe to monitor autophagy in conjunction with anti-ATG8/ATG18a antibody-Alexa Fluor 488. To indentify the effects of MeJA on autophagy, paraffin embedded sections with anti-ATG8/ATG18a antibody-Alexa Fluor 488 and MDC were prepared to examine the fluorescent intensity of colocalization.

The development of stem was divided into two stages: the primary growth stage (S1 and S2), the secondary growth stage (S3 and S4). In our previous reports, laticifers were differentiated from the cells outside the protophloem and accompanied with the development of stem. Laticifers were polygonal cells, which have thicker cell walls compared with peripheral cells. The fluorescence intensity of ATG8a and ATG18a was regularly changed with the development of the stem: weak (S1)-strong (S2, S3)-weak (S4) (Figure 6 and Figure 7). With the prolonging of MeJA treatment from 0 h, 36 h, to 48 h, the green fluorescence signals firstly became stronger (MeJA36) and then became weaker (MeJA48), but they still were stronger than that without treatment (CK). The change trend of the red fluorescence signals was consistent with that of green fluorescence signals, and could be well colocalized with green fluorescence signals, which indicated that ATG8 and ATG18a were colocalized with MDC to co-label autophagy structures. To show the trend of fluorescence intensity more directly, we quantified the merged fluorescence signals in laticifers, which was associated with autophagy structures. The change trend was showed in Figure 8, both ATG8 and ATG18 labeling presented differences (*p*-value ≤ 0.05) at MeJA 36 h (MeJA36) to the control in S1 developmental stages, and significant differences at MeJA 36 h (MeJA36) to the control (*p*-value ≤ 0.01) and differences at MeJA 48 h (MeJA48) to the control in S2 and S3 developmental stages. These results above suggested that MeJA could promote autophagy.

Consistent results were obtained in a TEM analysis. In our previous studies, the autophagosome formation in *E. kansui* laticifers has been reported [26]. As shown in Figure 9, autophagosomes which derived from ER dispersed in the cytoplasm (Figure 9A). The ER swelled into various shapes, crooked into cup-shaped structures (Figure 9B) and came into being double-membrane ring-like structures (Figure 9C), in which organelles and some cytoplasm cargos wrapped were being degraded. When the materials were treated with MeJA, the number of autophagosomes was significantly increased in laticifers at the same developmental stage (Figure 9D–F).

## 3. Discussion

At present, there are many reports about the effect of methyl jasmonate on development of laticifers, for example, inducing the biosynthesis of secondary plant metabolites and stimulating the formation and differentiation of milk ducts [11,12]. On this basis, comparative proteomics studies of latex in *E. kansui* laticifers treated with methyl jasmonate were conducted. After the treatment of methyl jasmonate, 509 proteins were obtained, among which 297 were annotated and 29 were significantly differentially expressed, they were related with phosphatidylinositol signaling system, starch and sucrose metabolism, pentose and glucuronate interconversions, ptoteasoe pathway, porphyrin and chlorophyll metabolism, galactose metabolism, amino sugar and nucleotide sugar metabolism, carbon metabolism, regulation of autophagy, glycan degradation, peroxisome pathway, purine metabolism, and endocytosis.

The well-established role of JA and its methyl ester MeJA in plant has been reported and various modes of MeJA action have been discussed [31]. MeJA signaling pathway interacts with other signaling pathways leading to transcriptional reprogramming in *Bupleurum kaoi* adventitious roots [32]. While it has been reported that MeJA induces ROS in cancer cells [23], we take this a step further, demonstrating that ROS also induces autophagy. Plant constantly sense and assess the ROS and reprogram their gene expression to respond to the changing conditions in their environment. ROS are metalloenzymes with three know isoforms (Mn-SOD, Fe-SOD, and Cu/Zn-SOD), depending on the active site mata confactor (Mn, Fe, or Gu/Zn), Cu/Zn-superoxide dismutase 2 (CSD2) have revealed a strong link between ROS and process such as growth, development and biotic and abiotic stress responses [33]. In our study, we found that Cu/Zn-superoxide dismutase was up-regulated under MeJA treatment (Figure 4). In general, it is accepted that ROS induced autophagy [34,35], autophagy, in turn, contributes to reduce oxidative damage and ROS levels through removal of protein aggregates and damaged organelles such as mitochondria [36,37,38]. ROS accumulation results in oxidative damage, which leads to mitochondrial dysfuction and cell injury [39,40]. Meanwhile, under these conditions, the autophagy which is characterized by the presence of autophagosome that engulfs cytosolic aged organelles and of autolysosome that degradates these damaged organelles is also induced [41,42]. In last decade, a large scale of studies has shown that ROS could initiate autophagosome formation and autophagic degradation acting as cellular signaling molecules [43,44].

Lysosomes play a central role in degradation of extracellular and intracellular macromolecules. These organelles contain hydrolytic enzymes capable of degrading proteins, proteoglycans, nucleic acids, and arid lipids. The autophagosome fuses with lysosomes to form autolysosomes, where their contents are then degraded by hydrolytic enzymes. In our previous study, it has been reported the autophagosomes fused another small acidic organelle that resembled lysosomes to form autolysosomes in *E. kansui* [26,27]. The acid α-l-fucosidase, a soluble component of lysosomes, is reported to involve in the hydrolytic degradation of fucose-containing molecules [45,46,47]. Lysosomal β-galactosidase, which catalyzes the hydrolysis of a variety of glycoconjugates with β-galactoside linkage, is widely distributed in living organisms [48]. Lysosomal cysteine protease is believed to be mainly involved in intracellular protein degradation [47]. Numerous researches about cysteine protease in latex have been reported, for example, cysteine proteases provide plants with a general defense mechanism and protein degradation during laticifer development [49,50,51]. In the present study, the accumulation of above proteins were all increased after MeJA treatment, therefore it was hypothesized that MeJA can promote autophagy in *E. kansui* laticifers. In order to further verify our hypothesis, the accumulation of ATG8 and ATG18a proteins and the expression of some autophagy related genes were performed by using methods of western blotting and qPCR respectively, which showed significant increasing trends under MeJA treatment 36 h (Figure 4, *p*-value ≤ 0.01,). In addition, we selected MDC, an acidic fluorescent drug, which was used as a probe to monitor lysosome; ATG8 and ATG18a, two important autophagy related proteins, which were used to monitor autophagosomes in conjunction with anti-ATG8/ATG18a antibody-Alexa Fluor 488 to detect the autophagy flow in laticifers under MeJA treatment by using laser confocal microscope. The results showed that the green and red fluorescence signals firstly became stronger (MeJA36) and then became weaker (MeJA48), but they still were stronger than that without treatment (the control). The quantified data of fluorescence signals of groups treated with MeJA presented differences (*p*-value ≤ 0.05) or significant differences (*p*-value ≤ 0.01) to the control during the laticifer development. In addition, TEM analysis on laticifers treated with MeJA 36 h showed that the number of autophagosomes was significantly increased compared to the control. These results above suggested that both the autophagy related protein and the lysosome protein were promoted under exogenous MeJA treatment, which induced the autophagy activity in the laticifers (Figure 6, Figure 7 and Figure 8).

Proteomics analysis, a powerful tool that can provide high-throughput information for the study of proteins involved in various biological processes. This technique has been widely employed in the researches of autophagy, for example, suberoylanilide hydroxamic acid (SAHA) stimulated autophagy in jurkat T-leukemia, Ras-driven cancer cells can up-regulated autophagy to support metabolism and macromolecular synthesis [43,52], and autophagy can mediated plant defense responses against Verticillium dahliae in *Arabidopsis* [53]. Based on the comparative iTRAQ protein profile and GO analyses, we characterized several functional associations between autophagy and MeJA treatment in *E. kansui*. Our data showed that autophagy was involved throughout the entire development of laticifers and some proteins that associated with autophagy were increased significantly when treated with MeJA. Furthermore, the autophagosome was increased when being treated with MeJA, indicating that autophagy in laticifers can promote plant defense responses against MeJA. In addition, we found that ROS protein was also increased in *E. kansui* laticifers under MeJA treatment. Some scholars had suggested that autophagy can be induced by H_2_O_2_ in *Arabidopsis* [18], which was produced through the ROS pathway. In mammalian cells, reactive oxygen species are essential for autophagy and specifically regulate the activity of Atg4 [54], and Atg8-PE is possibly protected from unregulated cleavage by Atg4 [55]. Recently, accumulating data have pointed to an essential role for ROS in the activation of autophagy [36,56]. As a result, these findings above suggest a hypothetical model, which showed that MeJA promotes autophagy through two possible ways in Figure 10. This model provides valuable information for subsequent research, which will be a very meaningful scientific problem to be solved in the future.

## 4. Materials and Methods

### 4.1. Plant Materials

*Euphorbia kansui* plants were grown in the field at the Botanical Garden of Northwest University in Shaanxi province (Shaanxi, People’s Republic of China). The plant materials at the same development stage were assigned to three groups (Two biological repeats per group), and sprayed with 0.5 mg/L MeJA (in Milli-Q water). The MeJA solution was sprayed as a fine mist to completely wet the adaxial side of each leaves. Latex samples from the groups treated with three time intervals (0, 36, and 48 h) were collected separately. The total RNA and total proteins of each group were immediately extracted for subsequent experiments. In the relevant text and figures, we used CK, MeJA36 and MeJA48 respectively to indicate the materials which is treated with MeJA for 0 h, 36 h, and 48 h.

### 4.2. Protein Preparation and iTRAQ Labeling

Latex protein extraction was prepared using a modified trichloroacetic acid (TCA)/acetone extraction method. In short, latex of *Euphorbia kansui* was mixed with equal volume of Tris-HCl (100 mM, pH 8.0), and centrifuged at 12,000 g for 10 min at 4 °C. Crude extract was dissolved in 5-fold volumes of ice-cold 10% (*v*/*v*) TCA in acetone (containing 0.07% β-mercaptoethanol), and stored overnight at −20 °C for precipitating the proteins out of the solution. Following centrifugation at 12,000 g for 30 min at 4 °C, this step repeated once, protein pellets were collected and dissolved in RIPA buffer (7 mol/L Urea, 2 mol/L Thiourea, 4% CHAPS, 2 mmol/L EDTA, 2 mmoL/L Tris, 1 mmoL/L PMSF). Proteins were reduced, alkylated, and subjected to tryptic hydrolysis. The tryptic peptide mixtures were collected and lyophilized for further analysis. The resulting lyophilized powders were re-dissolved in 1 M TEAB buffer, and then labeled with iTRAQ tags as follows: CK: iTRAQ tag 113 and 114, MeJA 36: iTRAQ tag 115 and 116, MeJA 48: iTRAQ tag 117 and 118. Follow being labeled, the iTRAQ reactions were incubated at room temperature for 2 h, and the peptide mixtures were combined and vacuum-dried completely. Each dried fraction of iTRAQ-labeled peptides was dissolved in mobile phase A (3% acetonitrile, 0.1% formic acid in deionized water) and performed strong cation exchange (SCX) on an Agilent 1200 HPLC with a gradient phase B (90% acetonitrile, 0.1% formic in deionized water). Based on UV absorbance at 210 nm and 280 nm, each SCX fraction was collected and vacuum-dried for LC-MS/MS analysis.

### 4.3. LC-MS/MS Analysis

The peptides were resuspended in phase A (3% acetonitrile, 0.1% formic acid in deionized water), then centrifuged at 20,000 g for 10 min. Subsequently, 2D nano-LC-MS/MS analyses were carried out using a nano-HPLC system (Agilent, Waldbronn, Germany), the supernatant was loaded onto a C18 precolumn, and then eluted onto analytical C18 column and packed inside it. The samples were then loaded onto a C18 reverse phase column in gradient phase B (90% acetonitrile, 0.1% formic in deionized water) for 60 min.

A triple TOF 5600 System, fitted with a Nanospray III source (AB SCIEX) and a pulled quartz tip as the emitter (New Objectives, MA), was performed for all measurements. Data was acquired using an ion spray voltage of 2.5 kV, curtain gas of 30 psi, nebulizer gas of 5 psi, the interface heater temperature of 150, mass spectra were acquired in an information dependent analysis mode. For IDA, survey scans were acquired within 250 milliseconds, and if they exceeded a threshold of 120 counts per second (counts/s), up to 35 product ion scans were collected. The total cycle time was fixed at 2.5 s. By monitoring a 40 GHz multichannel TDC detector with a four-anode channel detection, the four time bins per scan were summed at a pulser frequency value of 11 kHz. Coupled with iTRAQ adjust rolling collision energy, a normalized collision energy setting of 35 ± 5 eV, was applied to all precursor ions for collision-induced dissociation. Set dynamic exclusion to 1/2 of peak width (18 s), and then the precursor was refreshed to the exclusion list.

### 4.4. Protein Identification and Quantification

The proteins were identified based on the mRNA transcriptome of *Euphorbia kansui* in response to MeJA (NCBI Sequence Read Archive database, Accession number: SRP126436). To reduce the probability of false peptide identification, only peptides at the 95% confidence and a false discovery rate (FDR) less than 1% were qualified for further data analysis. Furthermore, identification of each confident protein involved one unique peptide at least. As for protein quantization, at least two unique peptides must contain in a protein. In Mascot, the quantitative protein ratios were weighted and normalized by the median ratio. The *p*-values were generated by ProteinPilot Sofware on the basis of the peptides used to quantify the respective proteins. *p*-value ≤ 0.05 were considered to be significantly differentially expressed proteins in normal and MeJA treated latex. At last, functional annotations of the proteins were conducted using Blast2GO program against the non-redundant protein database (NR; NCBI). The KEGG database (http://www.genome.jp/kegg/) was used to classify and group these identified proteins.

### 4.5. Bioinformatics Analysis

After proteins were identified, functional classifications, pathway and enrichment analyses of the GO on the differentially expressed proteins were analyzed using bioinformatics method. By using gene ontology analysis, proteins were classified according to three categories, including biological process, cell component, and molecular function, respectively. Pathway analyses of identified proteins were performed using the Kyoto Encyclopedia of Genes and Genomes (KEGG) database (http://www.genome.jp/kegg/). GO terms and KEGG pathways with corrected *p*-value ≤ 0.5 were considered significant.

### 4.6. Validation by Real-Time qPCR Analysis

Total RNA of latex in control group and MeJA treated group of *Euphorbia kansui* were isolated with a RNA extraction kit (Magen, China). The concentration and quality of each total RNA sample was measured by NanoDrop ND-2000 (Thermo, Waltham, USA) and 1% agarose gel. Approximately 5 μg of total RNA from each group was used for the first-strand cDNA synthesis using iscriptTM cDNA Synthesis Kit Real Time (Zoman, China). The cDNAs were kept at −20 °C. Real-time quantitative PCR analyses were performed by a two-step PCR procedure with 2 × SYBR Green qPCR Mix and the CFX96™ Real-Time PCR System (Bio-Rad). The real-time PCR mixture of 25 µL included 1 µL of cDNA solution, 12.5 µL of 2 × SBRY qPCR Mix, 0.5 µL of forward primer, and 0.5 µL of reverse primer, 10.5 µL of dd H2O water to final volume. PCR conditions were as follows: 94 °C for 2 min, followed by 40 cycles at 94 °C for 10 s and 60 °C for 30 s. Primes used for qRT-PCR was listed in the Appendix A. β-Actin was also amplified as an internal reference for the same sample and shown in Appendix A. The 2−ΔΔCt comparative CT method was used for calculating the genes expression.

### 4.7. Antibodies and Western Blot Analysis

Antibodies against ATG18a were generated with 6×His recombinant protein as antigen. Recombinant 6 × His-ATG18a proteins were purified from insoluble fractions as His fusion proteins using a Protein A column according to standard methods (GE) The sequence information of genes was listed in Appendix A. Following SDS-PAGE, gel pieces containing the recombinant proteins as identified by coomassie Brilliant Blue staining were extracted and injected directly into healthy rabbits. Preparation of protein samples was carried out as described previously (Fang et al. 2019). The protein concentration was measured by the BCA protein assay kit (Well Biotechnology Company, Changsha, China). A total of 50 μg of proteins of each group (CK, MeJA36, MeJA48) were mixed with sample buffer. Proteins were transferred onto nitrocellulose membranes using the Bio-Rad blotting system. The nitrocellulose membranes were then blocked with 3% BSA in TBS and 0.1% Tween-20 for at least 2 h at room temperature, followed by incubation overnight at 4 °C with anti-ATG8a antibody (dilution: 1/5000, Abcam, Massachusetts, USA) and anti-ATG18a antibody (dilutinnnon: 1/2000). After being washed thrice with TBST (10 min each), the membranes were incubated with horseradish peroxidase-labeled secondary antibody (dilution 1/2000, Abbline, California, US). After incubation finished, the membranes were washed three times (10 min each) and visualized using enhanced chemiluminescence detection system (ECL, Termo Fisher Scientifc, Shanghai, China) for 2 min before exposure to X-ray film. Western blotting bands were scanned and photographed with chemiluminescent gel imager (Tanon, Shanghai, China).

### 4.8. Confocal Immunofluorescence Studies

The developing vegetative apices of young stems of each group (CK, MeJA36, MeJA48) were treated as described previously. The samples were embedded in paraffin and sliced into 4 μm-thick sections. Before incubation with antibody, the samples were soaked in 10 mM citrate buffer (PH 6.0) at 95 °C three times for 10 min, then the sections were post-fixed in 4% paraformaldehyde for 20 min and treated with 1% Triton X-100 30 min, finally blocked in 3% bovine serum albumin (BSA) solution for 1 h at room temperature. After removing excess of BSA solution, the sections were incubated in anti-ATG8a antibody (dilution: 1:1000) overnight at 4 °C and 30 min at room temperature, and washed with PBST for three times. Subsequently, the sections were incubated with fluorescent isothiocyanate-conjugated goat anti-rabbit IgG (Alexa Fluor 488; Abcam) at a dilution of 1:1000 for 2.5 h, and 0.05 mM Monodansylcadaverine (MDC, Sigma) for 30 min respectively. It is noteworthy that the sections were washed with PBST for three times at each incubation interval. All confocal fluorescence images were collected using an Olympus FV1000 system. Finally, we quantified the mean fluorescence intensity in laticifers using the previous method [26].

### 4.9. Statistical Analysis

Data for continuous variables with normal distributions are presented as the mean and standard deviation (mean ± SD). Comparison between means was assessed by unpaired Student’s *t*-test and Statistical analyses were performed using SPSS10.0 software. Statistical significance was set at *p*-value ≤ 0.05. All assays were performed in triplicate.

### 4.10. Transmission Electron Microscopy

For transmission electron microscope (TEM) to visualize autophagy-related structures, samples from developing vegetative apices of young stems of the control group (CK) and MeJA treated group (MeJA36) were fixed in 2.0% paraformaldehyde and 2.5% glutaraldehyde in 0.1 M phosphate buffer (PH7.0) overnight at 4 °C. After washing in buffer, the sections were postfixed for 2 h at room temperature in buffered 1% OsO4, washed and dehydrated using a series of ethanol, and then embedded in Epon 812 (SPI, West Chester, PA, USA). Ultrathin sections of 70 nm were cut by an ultramicrotome (Leica, Wetzlar, Germany), stained with uranyl acetate and lead citrate, then examined and photographed under a TEM 300 (Philips, Amsterdam, The Netherlands). The quantitation of autophagosomes in TEM sections was performed following the method which described by Liu et al. [57].

## 5. Conclusions

Autophagy is an intracellular bulk degradation process, through which a portion of the cytoplasm is delivered to lysosomes to be degraded [15]. MeJA, a naturally occurring plant growth regulator, is known to regulate various plant development phenomena and response to environmental stress [1,2,3,4]. By using the *Euphorbia kansui* and MeJA as the research model, studies have been performed to investigate the effects of MeJA on autophagy. The current evidences confirmed that MeJA promotes autophagy in *E. kansui* laticifers. In addition, it was speculated that MeJA promotes autophagy through two possible ways: MeJA regulates the increase of ROS, which induces ATG8 accumulation, and MeJA also promotes ATG18 accumulation, they all result in the increase of autophagosome production. However, the specific mechanism should be further studied.

## Figures and Tables

**Figure 1 ijms-20-03770-f001:**
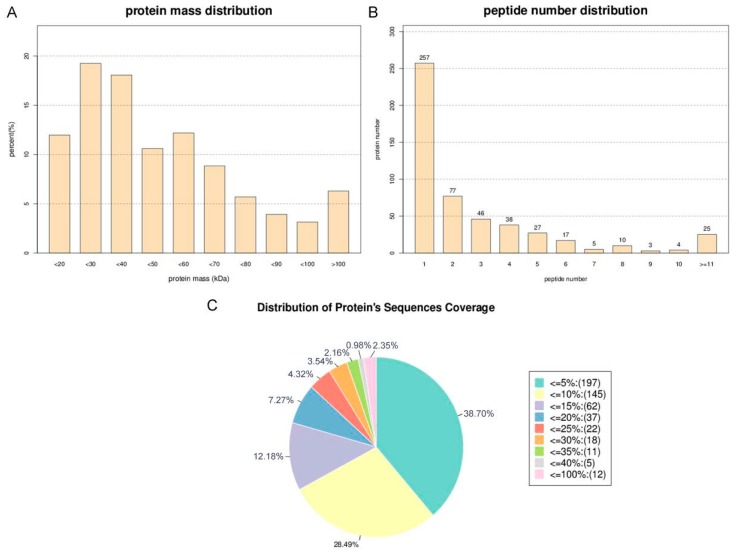
Identification and analysis of the *Euphorbia kansui* latex proteomics. (**A**) Identified proteins were grouped based on their protein mass. (**B**) The number of peptides that match to proteins as shown by Protein Piloy 5.0. (**C**) According to the protein sequence coverage, the identified proteins were classified into pie charts.

**Figure 2 ijms-20-03770-f002:**
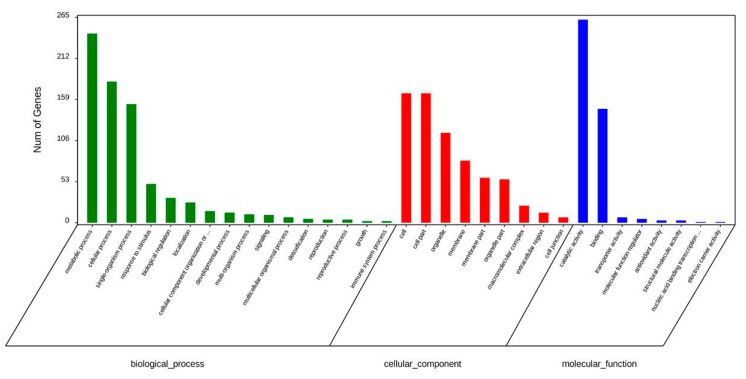
Gene ontology (GO) analysis of differentially accumulated proteins in *Euphorbia kansui* latex.

**Figure 3 ijms-20-03770-f003:**
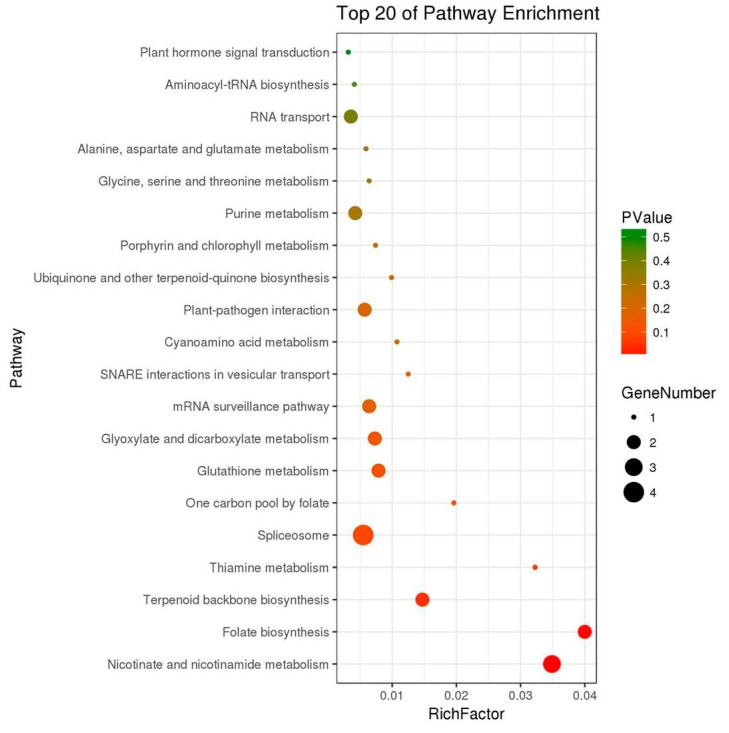
Kyoto Encyclopedia of Genes and Genomes (KEGG) pathway enrichment of proteins in *Euphorbia kansui* latex.

**Figure 4 ijms-20-03770-f004:**
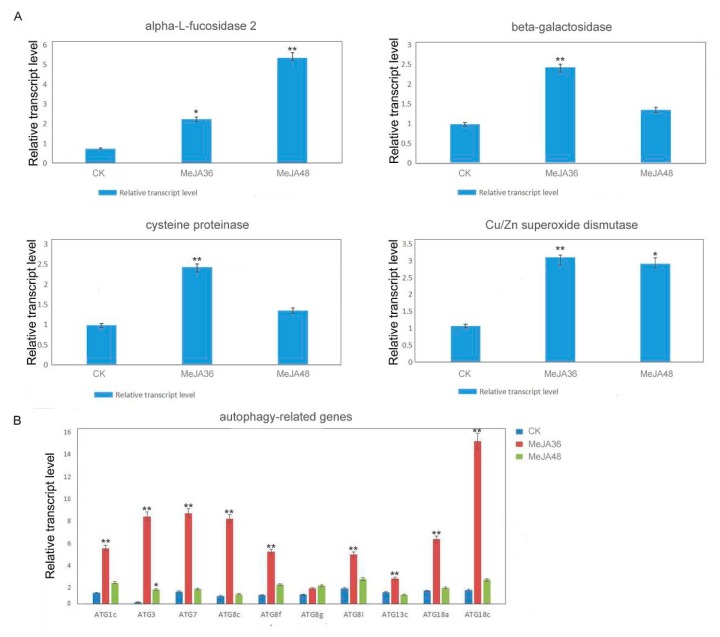
Transcriptional expression was significantly enhanced after being treated with Methyl jasmonate (MeJA). (**A**) qRT-PCR analysis of the transcription levels of selected genes that were significantly differentially expressed in latex proteomics. (**B**) Time-course gene expression profile of selected autophagy genes. “*” and “**” indicate statistically significant (*p*-value ≤ 0.05 or *p*-value ≤ 0.01 versus control groups), measured by the Student *t* test.

**Figure 5 ijms-20-03770-f005:**
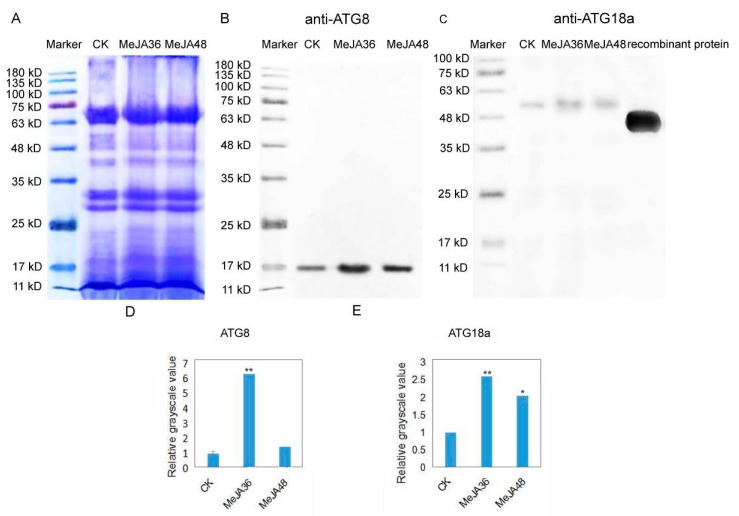
Western blot analysis of two autophagy related proteins in *Euphorbia kansui* laticifers of the control and MeJA treated groups. (**A**) Coomassie Brilliant Blue staining of the gel to show equal loading of proteins. (**B**,**C**) Western blot analysis of ATG8 and ATG18a. (**D**,**E**) Gray analysis of (**B**,**C**). “*” and “**” indicate statistically significant (*p*-value ≤ 0.05 or *p*-value ≤ 0.01 versus control groups), measured by the Student *t* test.

**Figure 6 ijms-20-03770-f006:**
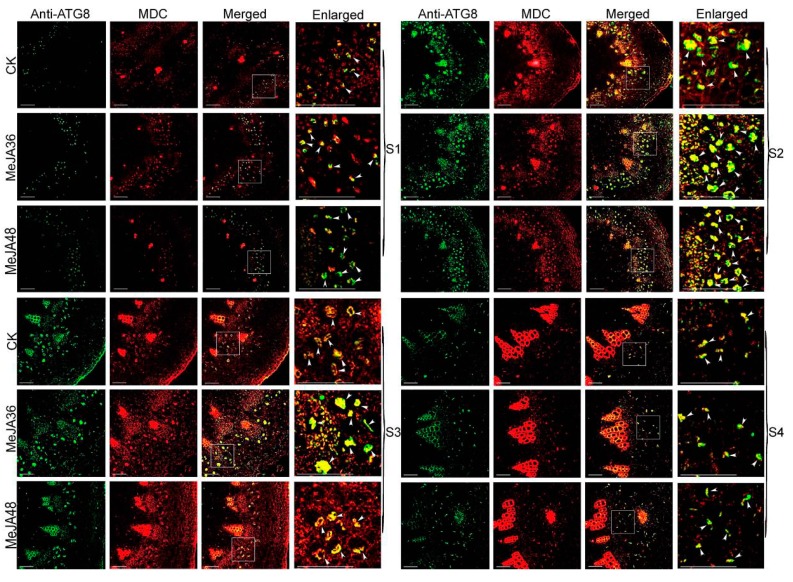
Autophagy activation monitoring by colocalization analysis of anti-ATG8 antibody-Alexa Fluor 488 and monodansylcadaverine (MDC) in laticifers of *Euphorbia kansui*. S1: the differentiation stage of the primary meristem of stems. S2, S3: the developing stage of the primary structure of stems. S4: the developing stage of the secondary structure of stems. Laticifers were depicted with white arrows. Bars = 100 nm.

**Figure 7 ijms-20-03770-f007:**
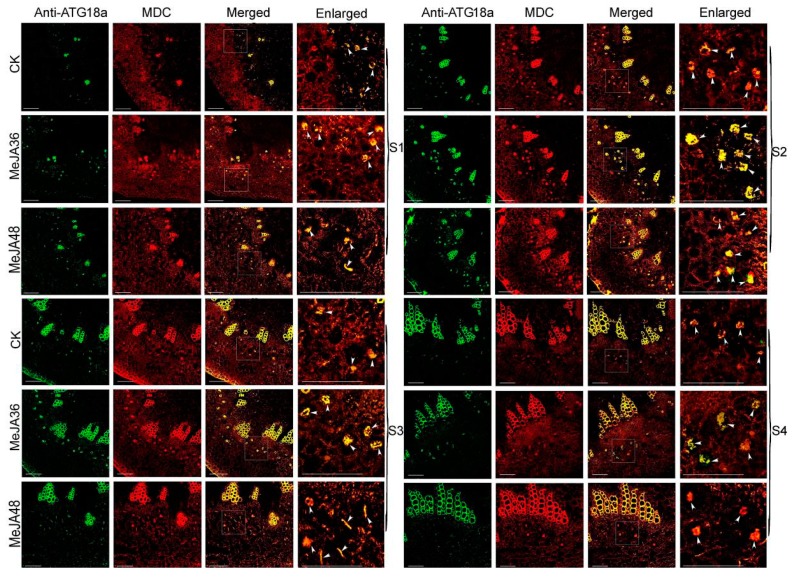
Autophagy activation monitoring by colocalization analysis of anti-ATG18a antibody-Alexa Fluor 488 and MDC in laticifers of *Euphorbia kansui*. S1: the differentiation stage of the primary meristem of stems. S2–S3: the developing stage of the primary structure of stems. S4: the developing stage of the secondary structure of stems. Laticifers were depicted with white arrows. Bars = 100 nm.

**Figure 8 ijms-20-03770-f008:**
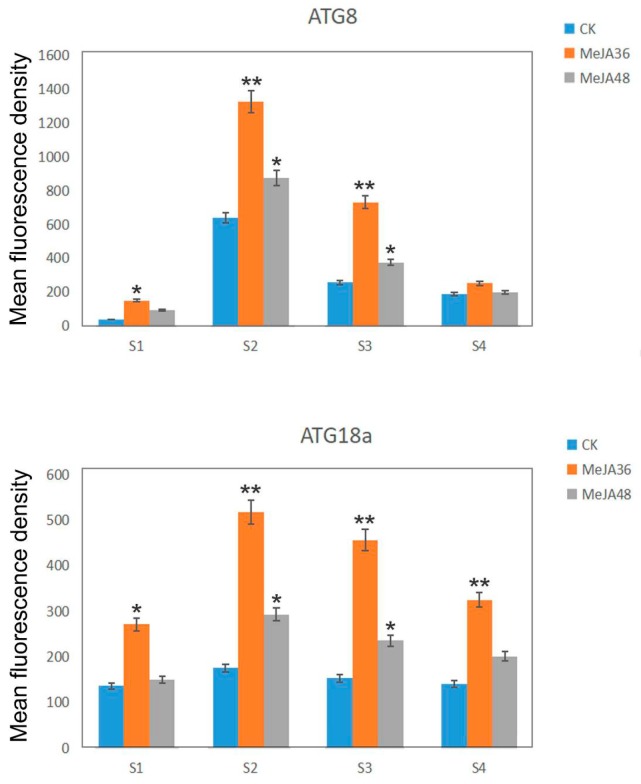
Quantifications of the mean fluorescence density of colocalization of ATG8 and ATG18a with MDC in laticifers of *Euphorbia kansui* at each development stage from S1 to S4. The bar diagram refers to the mean fluorescence density of the ATG8 and ATG18a associated autophagy structures. More than 150 laticifer cells for each stage were used for the quantification. Values represent the mean ± SE from three independent experiments. “*” and “**” indicate statistically significant (*p*-value ≤ 0.05 or *p*-value ≤ 0.01 versus control groups), measured by the Student *t* test.

**Figure 9 ijms-20-03770-f009:**
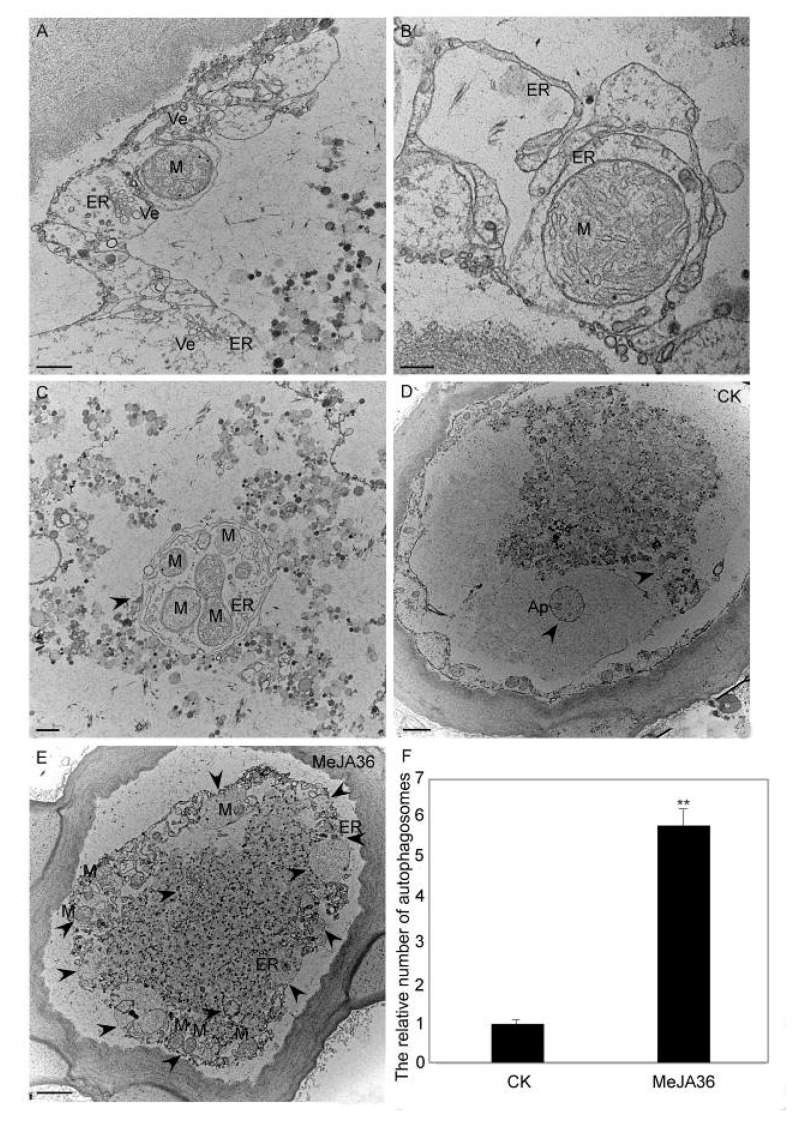
TEM analysis of the autophagy activation in laticifers of *Euphorbia kansui* treated with MeJA. (**A**) The early stages of autophagy in laticifers of *E. kansui*. (**B**) Autophagosomes are in the formative stage. (**C**) Representative TEM images of autophagy structures. (**D**) Laticifer TEM images of control groups of *E. kansui*. (**E**) Laticifer TEM images of *E. kansui* treated with MeJA. Arrows in (**B**–**E**) indicate autophagy structures. (**F**) Quantification of autophagy structures observed in laticifers of control groups and MeJA treated groups of *E. kansui*. Bar = 0.5 µm for (**A**,**C**), bar=0.2 µm for (**B**), Bar= 2 µm for (**D**,**E**). Mean and standard error (SE) were calculated from 10 visual fields at the same magnification. “**” indicate statistically significant (*p*-value ≤ 0.01 versus control groups), measured by the Student *t* test.

**Figure 10 ijms-20-03770-f010:**
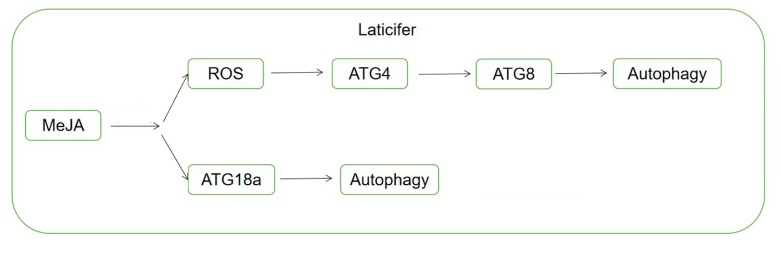
The hypothetical model of the mechanism of MeJA-mediated autophagy in *E. kansui* laticifer.

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
