# Peer review of "iTRAQ-Based Proteomics Analysis of Autophagy-Mediated Responses against MeJA in Laticifers of Euphorbia kansui L."

_ijms, 2019, doi:10.3390/ijms20153770_

Round 1

Reviewer 1 Report

The manuscript entitled “iTRAQ-based proteomics analysis of autophagy-mediated responses against MeJA in laticifers of Euphorbia kansui L” by Xiaoai Fang et al. is a well presented investigation of some aspects of MeJA-dependent responses as reflected by changes in the proteome of laticifers.
In my view the strength is that the authors present a solid sets of data, the weakness is that the “purpose of the the paper” remains vague and the scientific discussion/interpretation should be sharpened.
I propose that the authors state a hypothesis at the end of the Introduction and give the reader an idea what questions will be answered. This will remove some of the vagueness and give sharper direction to the arguments.

Major points:
- The term autophagy needs defining (including in the Abstract and Introduction). The relationship between apoptosis and autophagy also needs to be stated. This is particularly relevant, since it is a central part of the argument presented.
The same holds true for ATGa and ATG18a - to state in the Abstract that they are “important” is clearly insufficient, since all proteins are important.

- When I look at the literature, I note that autophagy is regulating cell death and that this process is dependent on salicylic acid (SA). There is obviously a link between SA, the immune response and senescence in A.t. (see e.g. The Plant Cell, Vol. 21: 2914–2927, September 2009). Given this link, I think the authors need to tell the reader what MeJA does (and possibly how) beyond general statements like “…. mediates diverse developmental process and defense responses which induce a variety of metabolites.”
- When the authors write “… and protecting cells against pathogen infection or other unfavourable conditions [15-21],” what do they mean. What type of pathogen (biotrophic or necrotrophic), and what sort of unfavourable conditions do they refer to. Incidentally refs. 15 to 21 are all over 10 year old.
- The authors do not really tell the reader why they work on Euphorbia kansui L. apart from the fact that it is a medicinal plant. To make this investigation plausible, the authors need to present a hypothesis that phrases why they chose their experimental system and what the reader can expect to learn. In other words, the Introduction must declare the scope of the works.
- Incidentally, the section that describes the biological/medicinal effects of  Euphorbia kansui L. is not properly referenced. What sort of extracts of the plant are used  “… [for] the treatment of edema, ascites, epilepsy and asthmatic cough…”and what is their effect? Are these statements based on scientific evidence (if so, we need to see citations), or clinical evidence based on trials with control groups?
In the Discussion, the authors write about the “well-established role of Jasmonic acid” instead of telling the reader what the role actually is, how it relates to e.g. ROS and ROS signaling. What is the link between JA, ROS and autophagy? I therefore suggest that the authors draw a Figure with the key autophagy pathways and the points of intersection with JA, ROS, MeJA etc.    
Minor points:
- The formatting (line adjustments) in lines 117 -129 need attention.
- the ref.  (dol: 10.1093/jxb/ery169) needs completing in the bibliography.

Author Response

Dear reviewer-in-Chief :

Thank you very much for your attention, the evaluation and comments on our manuscript "iTRAQ-based proteomics analysis of autophagy-mediated responses against MeJA in laticifers of Euphorbia kansui L.". These comments are valuable and very helpful for us to revise and improve our manuscript, as well as the important guiding significance to our researches. Meanwhile, we have revised the manuscript according to your kind advices . The responses to all comments are as follows.

Best regards

Sincerely yours Xia Cai

  Reviewer: 1

The manuscript entitled “iTRAQ-based proteomics analysis of autophagy-mediated responses against MeJA in laticifers of Euphorbia kansui L” by Xiaoai Fang et al. is a well presented investigation of some aspects of MeJA-dependent responses as reflected by changes in the proteome of laticifers.

In my view the strength is that the authors present a solid sets of data, the weakness is that the “purpose of the the paper” remains vague and the scientific discussion/interpretation should be sharpened.

I propose that the authors state a hypothesis at the end of the Introduction and give the reader an idea what questions will be answered. This will remove some of the vagueness and give sharper direction to the arguments.

Major points:

1- The term autophagy needs defining (including in the Abstract and Introduction). The relationship between apoptosis and autophagy also needs to be stated. This is particularly relevant, since it is a central part of the argument presented.

The same holds true for ATGa and ATG18a - to state in the Abstract that they are “important” is clearly insufficient, since all proteins are important.

Response: It has been modified in the paper: Page 1-2.

Page1 :

 Abstract Autophagy is a well-defined catabolic mechanism whereby cytoplasmic materials are engulfed into a structure termed the autophagosome. Methyl jasmonate (MeJA), a plant hormone, mediates diverse developmental process and defense responses which induce a variety of metabolites. In plants, little is known about autophagy-mediated responses against MeJA. In this study, we used high-throughput comparative proteomics to identify proteins of latex in the laticifers. The isobaric tags for relative and absolute quantification (iTRAQ) MS/MS proteomics were performed, and 298 proteins among MeJA treated groups and the control group of Euphorbia kansui were identified. It is interesting to note that 29 significant differentially expressed proteins were identified and their associations with autophagy and ROS pathway were verified for several selected proteins as follows: α-L-fucosidase, β-galactosidase, cysteine proteinase and Cu/Zn superoxide dismutase. Quantitative real-time PCR analysis of the selected genes confirmed the fact that MeJA might enhance the expression of some genes related to autophagy. The western blotting and immunofluorescence results of ATG8 and ATG18a which are two important proteins for autophagy the formation of autophagosomes also demonstrated that MeJA could promote autophagy at the protein level. Using the electron microscope, we observed an increase in autophagosomes after MeJA treatment. These results indicated that MeJA might promotes autophagy in E. kansui laticifers; and it was speculated that MeJA mediated autophagy through two possible ways: the increase of ROS induces ATG8 accumulation and then aotophagosome formation, and MeJA promotes ATG18 accumulation and then autophagosome formation. Taken together, our results provide several novel insights for understanding the mechanism between autophagy and MeJA treatment. However, the specific mechanism remains to be further studied in the future.

Page 2:

“Autophagy, a life-promoting lysosomal degradation pathway is conservative in eukaryotics [13,14]. Autophagy, which means “self-eating,”is a protein degradation process in which cells recycle cytoplasmic contents when subjected to environmental stress conditions or during certain stages of development [15]. This process mediates the degradation and recycling of cellular components through their segregation into double-membrane vesicles called autophagosomes, which will deliver the contents to lysosomes or vacuole for degradation by hydrolases. The functional relationship between apoptosis and autophagy is complex in the sense that, under certain circumstances, autophagy constitutes a stress adaptation that suppresses apoptosis, whereas in other cellular settings, it constitutes an alternative cell-death pathway [15]. In plant, the importance of autophagy has generated significant interest in various biological processes during normal growth and development, for example, protein aggregates are transported to vacuoles by a macroautophagic mechanism in nutrient-starved plant cells; autophagy maintains the metabolism and function of young and old stem cells [16,17]. Autophagy is essential for the degradation of oxidized proteins during oxidative stress in plants [18,19], for the degradation of aggregated proteins in nutrient-starved plant cells[15] and for limiting the cell death response during pathogen attack [20-22]. Num -independent non-apoptotic cell death in cancer [23-25], MeJA induces apoptosis via the ROS pathway, while simultaneously inducing pro-apoptotic autophagy in human non-small cell lung cancer [23]. However, few studies have been explored to elucidate the effect of MeJA on autophagy in plant, whether it stimulates autophagy via ROS pathway like mammalians, or promote autophagy directly? This problem needs to be solved.”

As for ATG8 and ATG18a, we explain their functions and importance in result 2.5 (Page 6 line 167-182)

“The autophagy related protein 8 (ATG8) is located on the autophagosome membrane and is often used as autophagosome marker in eukaryotic cells [27]. ATG8, whose activity was found to be essential for autophagosome formation, regulates mambrane elongation during autophagosome biogenesis [28]. The autophagy related protein 18 (ATG18) plays a role in the initial stage of autophagosome formation. After the cells receive the signal to initiate the autophagy process, phosphatidylinositol 3-kinase (PI3K) is responsible for the synthesis of phosphatidylinositol 3-phosphate, 1, 2-dipalmitoyl (PI3P) at the initial site of pre-autophagosome (PAS) membrane. ATG18a is located on the surface of autophagosome membrane by binding to PI3P and interacts with autophagy related protein 2 (ATG2) to form atg2-atg18a complex, whose role is circulating autophagy related protein 9 (ATG9) to transport lipids through PAS [28, 29]. Therefore, both ATG8 and ATG18a could be used to label the autophagosome structure.”

2- When I look at the literature, I note that autophagy is regulating cell death and that this process is dependent on salicylic acid (SA). There is obviously a link between SA, the immune response and senescence in A.t. (see e.g. The Plant Cell, Vol. 21: 2914–2927, September 2009). Given this link, I think the authors need to tell the reader what MeJA does (and possibly how) beyond general statements like “…. mediates diverse developmental process and defense responses which induce a variety of metabolites.”

Response: This is a good suggestion, which we have supplemented in the manuscript. (Page 13 line 312-332)

 Proteomics analysis, a powerful tool that can provide high-throughput information for the study of proteins involved in various biological processes. This technique has been widely employed in the researches of autophagy, for example, suberoylanilide hydroxamic acid (SAHA) stimulated autophagy in jurkat T-leukemia, Ras-driven cancer cells can up-regulated autophagy to support metabolism and macromolecular synthesis [43,52], and autophagy can mediated plant defense responses against Verticillium dahliae in Arabidopsis [53]. Based on the comparative iTRAQ protein profile and GO analyses, we characterized several functional associations between autophagy and MeJA treatment in E. kansui. Our data showed that autophagy was involved throughout the entire development of laticifers and some proteins that associated with autophagy were increased significantly when treated with MeJA. Furthermore, the autophagy autophagosome was increased when being treated with MeJA, indicating that autophagy was an important defense process in laticifers  in laticifers can promote plant defense responses against MeJA. In addition, we found that ROS protein was also increased in E. kansui laticifers under MeJA treatment.during the process. Some scholars had suggested that autophagy can be induced by H2O2 in Arabidopsis [54], which was produced through the ROS pathway. In mammalian cells, reactive oxygen species are essential for autophagy and specifically regulate the activity of Atg4 [55], and Atg8-PE is possibly protected from unregulated cleavage by Atg4 [56]. Recently, accumulating data have pointed to an essential role for ROS in the activation of autophagy [57,58]. As a result, these findings above suggest a hypothetical model, which showed that MeJA promotes autophagy through two possible ways in Fig. 10. This model provides valuable information for subsequent research, which will be a very meaningful scientific problem to be solved in the future.

3-When the authors write “… and protecting cells against pathogen infection or other unfavourable conditions [15-21],” what do they mean. What type of pathogen (biotrophic or necrotrophic), and what sort of unfavourable conditions do they refer to. Incidentally refs. 15 to 21 are all over 10 year old.

Response: It has been modified in the paper: Page 2 line 52-59, in addition, we have updated some recent references.

“In plant, the importance of autophagy has generated significant interest in various biological processes during normal growth and development, for example, protein aggregates are transported to vacuoles by a macroautophagic mechanism in nutrient-starved plant cells; autophagy maintains the metabolism and function of young and old stem cells [16,17]. Autophagy is essential for the degradation of oxidized proteins during oxidative stress in plants [18,19], for the degradation of aggregated proteins in nutrient-starved plant cells[15] and for limiting the cell death response during pathogen attack [20-22].

Refs:

              16. Toyooka, K.; Moriyasu, Y.; Goto, Y.; Takeuchi, M.; Fukuda, H.; Matsuoka, K. Protein            aggregates are transported to vacuoles by a macroautophagic mechanism in nutrient-starved plant cells. Autophagy 2006, 2, 96-106.

17. Ho, T.T.; Warr, M.R.; Adelman, E.R.; Lansinger, O.M.; Flach, J.; Verovskaya, E.V. Autophagy maintains the metabolism and function of young and old stem cells. Nature. 2017, 543, 205-210.

18. Xiong, Y.; Contento, A.L.; Nguyen, P.Q.; Bassham, D.C. Degradation of oxidized proteins by autophagy during oxidative stress in Arabidopsis. Plant Physiol. 2007, 143, 291-9.

19. Shin, J.H.; Yoshimoto, K.; Ohsumi, Y.; Jeon, J.S.; An, G. OsATG10b, an autophagosome component, is needed for cell survival against oxidative stresses in rice. Mol Cells 2009, 27, 67-74.

20. Zheng, P.; Wu, J.; Sahu, S. K.; Zeng, H.; Huang, L.; Liu, Z. Loss of alkaline ceramidase inhibits autophagy in Arabidopsis and plays an important role during environmental stress response. Plant Cell Environ. 2018, 41, 4.

21. Hayward, A.P.; Dinesh-Kumar, S.P. What can plant autophagy do for an innate immune response?. Annu Rev Phytophatholo. 2011, 49, 557-576.

22. Wang, Y.; Zhou, J.; Jingquan, Y. U. The critical role of autophagy in plant responses to abiotic stresses. Front Agri Sci Eng. 2017, 4.

3- The authors do not really tell the reader why they work on Euphorbia kansui L. apart from the fact that it is a medicinal plant. To make this investigation plausible, the authors need to present a hypothesis that phrases why they chose their experimental system and what the reader can expect to learn. In other words, the Introduction must declare the scope of the works.

Response: It has been modified in the paper: Page 2 line 76-82.

 In various species, laticifers are mainly the synthesis and storage sites of secondary metabolites, which have important economic value. The research of laticifer development is of great significance to further study the regulation and synthesis mechanism of important secondary metabolites. The non-articulated laticifers develop from a single primary cell; as a result, the latex is its protoplasm apart from the cell wall and plasma membrane. Therefore, latex from non-articulated laticifers in E. kansui is the best material for proteomic research to analyses the laticifer development and secondary metabolite synthesis.

4- Incidentally, the section that describes the biological/medicinal effects of  Euphorbia kansui L. is not properly referenced. What sort of extracts of the plant are used  “… [for] the treatment of edema, ascites, epilepsy and asthmatic cough…”and what is their effect? Are these statements based on scientific evidence (if so, we need to see citations), or clinical evidence based on trials with control groups?

Response: In order to make our manuscript better understood, we have deleted this section, and  updated in the paper: Page 2 line 71-82.

Euphorbia kansui L. is a medicinal plant belonging to the Euphorbiaceae family, which is well known for its milky latex substance. The roots of E. kansui   are regularly used as traditional medicine for the treatment of edema, ascites, epilepsy and asthmatic cough. The latex is also used to promote wound healing. The latex occurs within specialized secretory cells that are called as laticifers. The occurrence of non-articulated laticifers has been noted in the literatures for the Euphorbiaceae family. In various species, laticifers are mainly the synthesis and storage sites of secondary metabolites, which have important economic value. The research of laticifer development is of great significance to further study the regulation and synthesis mechanism of important secondary metabolites. The non-articulated laticifers develop from a single primary cell; as a result, the latex is its protoplasm apart from the cell wall and plasma membrane. Therefore, latex from non-articulated laticifers in E. kansui is the best material for proteomic research to analyses the laticifer development and secondary metabolite synthesis.

5- In the Discussion, the authors write about the “well-established role of Jasmonic acid” instead of telling the reader what the role actually is, how it relates to e.g. ROS and ROS signaling. What is the link between JA, ROS and autophagy? I therefore suggest that the authors draw a Figure with the key autophagy pathways and the points of intersection with JA, ROS, MeJA etc.    

Response: Thank you very much for your good suggestion, so that our manuscript can be better understood by readers. As for the relationship between MeJA, ROS and autophagy, we elaborated on it in the introduction and supplemented it in the discussion section. 

Page 13 line 264-270

“The well-established role of Jasmonic acid (Ja) and its methyl ester (methyl jasmonate, MeJA) in plant has been reported and various modes of MeJA action have been discussed [31]. MeJA signaling pathway interacts with other signaling pathways leading to transcriptional reprogramming in Bupleurum kaoi adventitious roots [32]. While it has been reported that MeJA induces ROS in cancer cells [23], we take this a step further, demonstrating that ROS also induces autophagy. Plant constantly sense and assess the ROS and reprogram their gene expression to respond to the changing conditions in their environment. ”

Page 13 line 325-332 

Some scholars had suggested that autophagy can be induced by H2O2 in Arabidopsis [54], which was produced through the ROS pathway. In mammalian cells, reactive oxygen species are essential for autophagy and specifically regulate the activity of Atg4 [55], and Atg8-PE is possibly protected from unregulated cleavage by Atg4 [56]. Recently, accumulating data have pointed to an essential role for ROS in the activation of autophagy [57,58]. As a result, these findings above suggest a hypothetical model, which showed that MeJA promotes autophagy through two possible ways in Fig. 10. This model provides valuable information for subsequent research, which will be a very meaningful scientific problem to be solved in the future.

 Figure 10. The hypothetical model of the mechanism of MeJA-mediated autophagy in E. kansui laticifer.

Minor points:

1-The formatting (line adjustments) in lines 117 -129 need attention.

Response: The formatting (line adjustments) in lines 117 -129 have been modified. (Page 5 lines 145-158)

2- the ref. (dol: 10.1093/jxb/ery169) needs completing in the bibliography.

Response: The ref. (dol: 10.1093/jxb/ery169) has been modified

“Deng, X.; Guo, D.; Yang, S.; Shi, M.; Chao, J.; Peng, S.; Tian, W. Jasmonate signalling in regulation of rubber biosynthesis in laticifer cells of rubber tree (Hevea brasiliensis Muell. Arg.). J Exp Bot. 2018, 69: 3559-3571

Reviewer 2 Report

This manuscript is applied proteomic analysis on autophagy responses of Euphorbia kansui L. against MeJA hormone. The idea and concept of the study is not novel but still valuable to understand the mechanism of plant response to environment factors. Thing is that many proteins were identified however only 1 Cu/Zn superoxide dismutase related to ROS was analysed. In particular the levels of many ATG genes were up-regulated after 36 h but only 2 ATG8 and ATG18a (not the most expression levels) were used for in-dept analysis. I am not sure the values are meaningful or not. Based on the above reasons, it is difficult to be accepted for publication in the IJMS as it is.

1. “These results indicated that MeJA might promote autophagy via ROS, or enhance autophagy directly??”. The results could or couldn’t answer the research questions is “been explored to elucidate the effect of MeJA on autophagy in plant, whether it stimulates autophagy via ROS pathway like mammalians, or promote autophagy directly? This problem needs to be solved.” If so, authors need to claim it in abstract.

2. Number the table according to the order of first appearance in the text. Please check again in manuscript.

3. The percentage of identified proteins are not clear in Fig 1C. Please clarify.

4. Please clarify difference of three groups used in this study.

5. Figure 4, CK, T1, T2 present for which group? The legend should be removed. I am not sure the statistical analysis values are meaningful or not. It seem significant difference in cysteine proteinase and beta-galactosidase between CK and T2. Also, in ATG1c, ATG8f, ATG8g, ATG8i between 0h and 48h.

6. Why ATG8 was not shown in Figure 4 but was selected in immunoblotting analysis?

7. Also these 2 genes ATG8 and ATG18a was not mentioned as main expression in previous results. Why Genes of ATG1c, ATG3, ATG7, ATG8s and ATG18s not be used for further study?

8. In separate conclusion section are needed to add as the last paragraph.

Euphorbia kansui and E. kansui should be italic

p-velue à p-value

Author Response

Dear reviewer-in-Chief :

Thank you very much for your attention, the evaluation and comments on our manuscript "iTRAQ-based proteomics analysis of autophagy-mediated responses against MeJA in laticifers of Euphorbia kansui L.". These comments are valuable and very helpful for us to revise and improve our manuscript, as well as the important guiding significance to our researches. Meanwhile, we have revised the manuscript according to your kind advices . The responses to all comments are as follows.

Best regards

Sincerely yours Xia Cai

Review 2:

This manuscript is applied proteomic analysis on autophagy responses of Euphorbia kansui L. against MeJA hormone. The idea and concept of the study is not novel but still valuable to understand the mechanism of plant response to environment factors. Thing is that many proteins were identified however only 1 Cu/Zn superoxide dismutase related to ROS was analysed. In particular the levels of many ATG genes were up-regulated after 36 h but only 2 ATG8 and ATG18a (not the most expression levels) were used for in-dept analysis. I am not sure the values are meaningful or not. Based on the above reasons, it is difficult to be accepted for publication in the IJMS as it is.

1. “These results indicated that MeJA might promote autophagy via ROS, or enhance autophagy directly??”. The results could or couldn’t answer the research questions is “been explored to elucidate the effect of MeJA on autophagy in plant, whether it stimulates autophagy via ROS pathway like mammalians, or promote autophagy directly? This problem needs to be solved.” If so, authors need to claim it in abstract.

Response:  It has been modified in the paper: Page 1 lines26-28.

“Taken together, our results provide several novel insights for understanding the mechanism between autophagy and MeJA treatment. However, the specific mechanism remains to be further studied in the future. “

2.Number the table according to the order of first appearance in the text. Please check again in manuscript.

Response: We have checked the tables and renumbered them.

Supplementary Materials: Table S1 All the latex proteins identified in Euphorbia kansui. Table S2 Significantly differentially expressed proteins in Euphorbia kansui latex treated with MeJA. Table S3 The KEGG pathways of proteins in Euphorbia kansui latex. Table S4 Primers for qRT-PCR. Table S5 Antibody information.

3. The percentage of identified proteins are not clear in Fig 1C. Please clarify.

Response: We have clarified the percentage of identified proteins in Fig.1C.

4. Please clarify difference of three groups used in this study.

Response: We have clarified difference of three groups used in this study. (Page 13 line 343-345.)

“ In the relevant text and figures, we used CK, MeJA36 and MeJA48 respectively to indicate the materials which is treated with MeJA for 0 h, 36 h and 48 h.

5.Figure 4, CK, T1, T2 present for which group? The legend should be removed. I am not sure the statistical analysis values are meaningful or not. It seem significant difference in cysteine proteinase and beta-galactosidase between CK and T2. Also, in ATG1c, ATG8f, ATG8g, ATG8i between 0h and 48h.

Response: We have modified the Figure 4.

6.Why ATG8 was not shown in Figure 4 but was selected in immunoblotting analysis?

Response: In Figure 4, some of the ATG8 homologous genes were tested, like ATG8c, f, g, i. The autophagy related protein 8 (ATG8) is located on the autophagosome membrane and is often used as autophagosome marker in eukaryotic cells, so we use it to detect autophagy.

Also these 2 genes ATG8 and ATG18a was not mentioned as main expression in previous results. ATG1c, ATG3 and ATG7 are involved in the regulation of autophagy, while ATG8 is involved in the formation of autophagosomes and distributed on the membrane of autophagosome, and is widely used as the marker of autophagosome at present. Therefore, we use the anti-ATG8 antibody to detect autophagy. In addition, although we found more than 30 autophagy related genes (atgs) in E. kansui transcription database, we only detected ATG18a in the latex proteome database. This indicated that ATG18a is more abundantly accumulated in laticifers than the other ATGs,  so we cloned ATG18a and prepared its antibodies to support our research. More importantly, these two proteins may be involved in the mechanism of MeJA-mediated autophagy. We elaborate on them in our discussion.

Page 13 line 325-332

“Some scholars had suggested that autophagy can be induced by H2O2 in Arabidopsis [54], which was produced through the ROS pathway. In mammalian cells, reactive oxygen species are essential for autophagy and specifically regulate the activity of Atg4 [55], and Atg8-PE is possibly protected from unregulated cleavage by Atg4 [56]. Recently, accumulating data have pointed to an essential role for ROS in the activation of autophagy [57,58]. As a result, these findings above suggest a hypothetical model, which showed that MeJA promotes autophagy through two possible ways in Fig. 10. This model provides valuable information for subsequent research, which will be a very meaningful scientific problem to be solved in the future.”

 Figure 10. The hypothetical model of the mechanism of MeJA-mediated autophagy in E. kansui laticifer.

7.In separate conclusion section are needed to add as the last paragraph.

Response: The conclusion section has been added in page 16.

“Conclusions

Autophagy is an intracellular bulk degradation process, through which a portion of the cytoplasm is delivered to lysosomes to be degraded [15]. MeJA, a naturally occurring plant growth regulator, is known to regulate various plant development phenomena and response to environmental stress [1-4]. By using the Euphorbia kansui and MeJA as the research model, studies have been performed to investigate the effects of MeJA on autophagy. The current evidences confirmed that MeJA promotes autophagy in E. kansui laticifers. In addition, it was speculated that MeJA promotes autophagy through two possible ways: MeJA regulates the increase of ROS, which induces ATG8 accumulation, and MeJA also promotes ATG18 accumulation, they all result in the increase of autophagosome production. However, the specific mechanism should be further studied.

8.Euphorbia kansui and E. kansui should be italic

p-velue à p-value

Response: These mistakes all have been modified.

Round 2

Reviewer 2 Report

Dear Authors,
Thank you for your careful response to the reviewer's comments. I think the manuscript can be accepted at the current stage.